# Improved Particle Swarm Path Planning Algorithm with Multi-Factor Coupling in Forest Fire Spread Scenarios

**Kaiyi Lin** [1,2], **Lifan Zhang** [3], **Lida Huang** [1,*] , **Zhili Feng** [3] **and Tao Chen** [1,*]

1 Institute of Public Safety Research, Department of Engineering Physics, Tsinghua University, Beijing 100084, China; linky21@mails.tsinghua.edu.cn
2 Shenzhen International Graduate School, Tsinghua University, Shenzhen 518071, China
3 Beijing Global Safety Technology Co., Ltd., Beijing 100094, China
* Correspondence: hld999@yeah.net (L.H.); chentao.b@tsinghua.edu.cn (T.C.)

**Abstract:** In this paper, a solution based on an improved particle swarm algorithm is proposed for the path planning problem without a road network in forest fire rescue scenarios. The algorithm adopts an adaptive inertia weight and a dynamically updated learning factor strategy to enhance the global and local search capabilities of the algorithm. In terms of cost function design, the article considers three factors: path length, terrain slope, and obstacle avoidance ability to ensure the safety and effectiveness of the path. The experimental results show that: (1) the path planning algorithm based on improved particle swarm optimization can effectively avoid spreading wildfire and reach the designated target point with a good "detour" effect; (2) the path planned by the improved PSO algorithm performs better than the original PSO algorithm in terms of fitness evaluation and average slope; and (3) changes in the particle population, dimensions, and learning factors in the particle swarm optimization algorithm can affect the convergence of the final path. Increasing the particle dimensions can bring more reasonable and specific paths; decreasing the learning factor increases the convergence iterations, but also obtains a better path planning solution and higher fitness.

**Keywords:** forest fire spreading; path planning; particle swarm optimization





## 1. Introduction

In recent years, forest fires have become a frequent occurrence in many countries worldwide due to extreme weather conditions, which pose a serious threat to human life and property safety. For example, forest fires in California in 2018 resulted in direct and indirect economic losses of around USD 14.85 billion, accounting for 9.1% of the state's annual GDP [1]. In Australia, the massive forest fires in 2020 caused 28 deaths and the loss of three billion animals [2]. Moreover, in Turkey, from 2000 to 2020, approximately 63,724 forest fires occurred, destroying roughly 320,000 hectares of forest [3]. As forest fires typically take place in remote and steep mountainous areas, they significantly impact the natural environment [4]. Additionally, forest fires tend to spread and grow larger due to various factors such as difficult accessibility and fire and smoke diffusion. These factors make it challenging for firefighters to find suitable routes for firefighting efforts. Therefore, obtaining a reasonable path that avoids forest fire risks and meets the needs of firefighting operations is of great significance when facing complex dynamic fire environments.

Contemporary research in path planning mainly focuses on unmanned aerial vehicles (UAVs) designed for aerial navigation. Wang et al. proposed an optimal path planning method for forest fire rescue UAVs using a vortex search algorithm that factors in two important considerations: spatial terrain and UAV energy [5]. Huo et al. proposed a task assignment and path planning method for multiple UAVs, developing a planning model that considers time resolution and balance constraints [6]. Path planning is also utilized in forest fire fighting decision-making. For instance, Sakellariou et al. developed a spatial decision support system (SDSS) that instructs the closest fire truck to the affected

area by finding the most efficient route. [7]. However, these path planning studies are seldom carried out in dynamic forest fire zones that do not have any road network. For path planning in complex dynamic environments, swarm intelligence algorithms are a good solution: Soot et al. combined the cuckoo search (CS) algorithm, bat algorithm (BA), and firefly algorithm (FA) in the field of path planning [8]; Ghamry et al. utilized particle swarm optimization (PSO) to create a path plan for multiple UAVs on forest fire missions [9]. From these studies, it is apparent that the efficiency and robustness exhibited by swarm intelligence algorithms while addressing complex dynamic challenges make them suitable for forest fire path planning scenarios that have numerous coupled risks. Among swarm intelligence algorithms, the most notable is the particle swarm optimization (PSO) algorithm, recommended by Kennedy and Eberhart in 1995, which serves a critical role in swarm intelligence computing. Initially, this algorithm was used to mimic the social behaviors of bird flocks and fish schools, and later it was further applied to optimization problems. Numerous improved versions of this method have been utilized to address a variety of complex engineering and scientific optimization challenges, such as power systems [10], mechanical engineering [11], image processing [12], data mining [13], and cloud computing [14]. Compared with other heuristic optimization algorithms, PSO has several advantages, including fewer adjustable parameters, the ability to handle target functions with stochastic properties, and the ability to conduct iterative calculations without requiring a good initial solution. Nonetheless, the traditional PSO algorithm has some restrictions, such as a reduction in population diversity with the increase in iteration times, and the likelihood of being caught in local optima.

In the present study, it was observed that most path planning algorithms are primarily designed for drones, whereas path planning for ground fires is predominantly conducted in enclosed fire areas such as buildings [15,16]. Despite this, only a handful of studies have investigated path planning specifically for complex environments, such as wildland fire spread across terrains of varying slopes and forest vegetation. The challenges associated with path planning research in such situations mainly arise from the difficulty of analyzing and modeling dynamic simulation data for wildfires, as well as the inability to effectively characterize the search space. In addition, the large variability in search space in path planning often leads to insufficient algorithm efficiency. Our research endeavored to address these issues with the following improvements and innovations: firstly, we employed a topological vertical partitioning method based on topology to model complex dynamic forest fire spread environments, thereby reducing the complexity of path planning in dynamic space based on the characteristics of fire spread dynamics; secondly, we introduced adaptive inertia weight and an adaptive learning factor calculation method to solve the issue of the particle swarm optimization algorithm being prone to fall into local optimal solution in path planning; finally, considered the dynamic nature of wildfires, including slope changes and spatiotemporal spread, as path planning scenarios in the application. We incorporated slope cost, personnel speed, and fire range as optimization objectives, ensuring the feasibility and applicability of planned paths.

The article is structured as follows. Section 1 elucidates the significance of path planning in forest fire scenarios and provides an overview of the existing research. Section 2 introduces the topological vertical partitioning method and the improved particle swarm path planning algorithm used in this study. Section 3 deploys the simulation results, demonstrating the optimization effects of the algorithm under varying parameters. Section 4 presents a summary of the paper and highlights future research directions for path planning algorithms in forest fire scenarios.

## 2. Materials and Methods

The path planning problem for forest fire fighting scenarios involves two subjects: the forest fire rescue team and the spreading fire. To model these mathematically, the algorithm in this study utilizes the configuration space. The algorithm's overall design concept is illustrated in Figure 1.

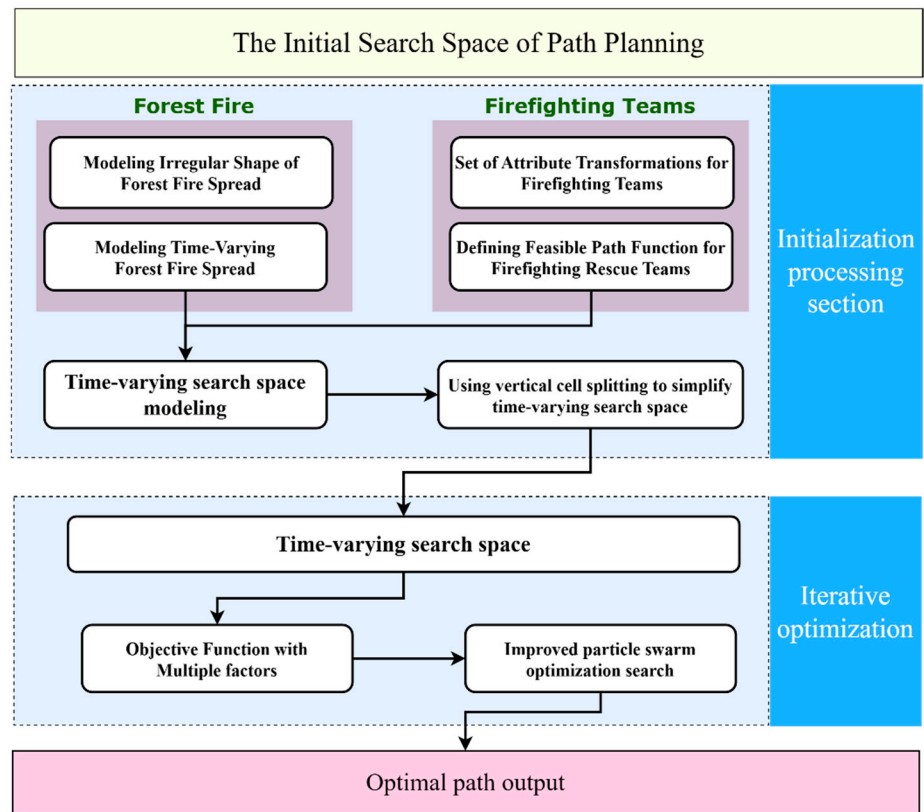

**Figure 1.** The overall design of the improved particle swarm optimization algorithm for path planning.

The algorithm comprises two main components. The first part is to numerically model the physical space that necessitates path planning. This involves capturing the initial configuration of the forest fire rescue team and the initial state of the forest fire field, while tracking their evolution over time to generate a time-varying search space. The second component is the optimization of the generated time-varying search space using an improved particle swarm method based on the prescribed cost function of the planned path. We provide a detailed account of the mathematical modeling process and path optimization process for both in the following sections.

### 2.1. Spatial Modeling of Forest Fire Spread

The spread of forest fires is characterized by an irregular shape influenced by many factors, such as combustible materials, temperature, wind speed, and wind direction in the forested region. To represent such irregular shapes, two strategies are typically used: (1) boundary representation and (2) solid representation [17]. In this study, the three-dimensional forest fire spreading scene defines $\mathcal{W} = \mathbb{R}^3$ as the three-dimensional state space for path planning, whereas solid representation establishes the forest fire spreading space, $\mathbb{F}$. For space, $\mathbb{F}$, we assume that the fire field is a convex polyhedron, $\mathbb{D}$, which is composed of vertices, edges, and faces. Each edge serves as a boundary between two faces, whereas each vertex forms the perimeter between three or more edges. Finally, each face has at least three vertices. The equation of planes passing through these points takes the following form:

$$ax + by + cz + d = 0 \tag{1}$$

in which the constants $a, b, c, d \in \mathbb{R}$.

Construct $f : \mathbb{R}^3 \to \mathbb{R}$, $f(x, y, z) = ax + by + cz + d$. Let $m$ be the number of faces. For each face of $\mathbb{D}$, a half-space $\mathbb{H}_i$ is defined as a subset of $\mathcal{W}$:

$$\mathbb{H}_i = \{(x, y, z) \in \mathcal{W} | f_i(x, y, z) \leq 0\} \tag{2}$$

Thus, a convex polyhedron is defined as the intersection of a finite number of half-spaces, as follows:

$$\mathbb{D} = \mathbb{H}_1 \cap \mathbb{H}_2 \cap \ldots \cap \mathbb{H}_m \tag{3}$$

Furthermore, the forest fire spread space, $\mathbb{F}$, is defined as:

$$\mathbb{F} = \mathbb{D}_1 \cup \mathbb{D}_2 \cup \ldots \cup \mathbb{D}_m \tag{4}$$

### 2.2. State Space of Firefighting Rescue Configuration Modeling

Defining the set of rescue team transformations is essential for modeling rescue team movement. This set encompasses future transformed versions of attributes, such as rescue team position, state, and direction, at a specific point in time. The collection of all future attribute transformations of the rescue team is referred to as the configuration space, $\mathcal{C}$ [18]. Paths between points in the configuration space are continuous and determine whether or not two points are connected to each other.

$$X = \tau(s) = \begin{cases} 0, & \text{path blocking} \\ 1, & \text{path connected} \end{cases} \tag{5}$$

The visited states of the points in $\tau(s)$ are denoted by $x_1, x_2, \ldots$, where $s \in [0, 1]$. In addition, the connectivity of the path is determined as follows:

A topological space, $X$, is considered connected if, for all $x_1, x_2 \in X$, there exists a path such that $\tau(0) = x_1$ and $\tau(1) = x_2$.

### 2.3. Time-Varying Search Space Modeling

To precisely model the forest fire spread space, $\mathbb{F}$, the boundary function, $f$, of the convex polyhedron is defined as a linear function. In this context, to more accurately determine the spread of forest fire over time, the time variable is introduced into this function, as follows:

$$\mathbb{H}_i^t = \{(x, y, z, t) \in \mathcal{W} | f_i(x, y, z, t) \le 0\} \tag{6}$$

where $(x, y, z, t)$ represents the coordinates of the fire front during the forest fire spread at time $t$. Additionally, we can define the time-varying fire field, $\mathbb{D}$, and the time-varying forest fire spread space, $\mathbb{F}^t$, as follows.

$$\mathbb{D}^t = \mathbb{H}_1^t \cap \mathbb{H}_2^t \cap \ldots \cap \mathbb{H}_m^t \tag{7}$$

$$\mathbb{F}^t = \mathbb{D}_1^t \cup \mathbb{D}_1^t \cup \ldots \cup \mathbb{D}_1^t \tag{8}$$

The time-varying path search space, $\mathcal{C}_{search}^t$, is established by defining the time-varying state forest fire spread space, $\mathbb{F}^t$. To obtain $\mathcal{C}_{search}^t$, we compute the path obstacle space, $\mathcal{C}_{obs}^t$, from the fire rescue team's configuration space, $\mathcal{C}$. The complement of $\mathcal{C}_{obs}^t$ in $\mathcal{C}$ yields:

$$\mathcal{C}_{obs}^t = \{q \in \mathcal{C} | A(q) \cap \mathbb{F}^t \ne \varnothing\} \tag{9}$$

where $q \in \mathcal{C}$ represents the configuration of rescue team A, with $q = (x_t, y_t, z_t, h)$, where $h$ denotes the unit quaternion. Additionally, $\mathcal{C}_{obs}^t$ refers to the set of all configurations, $q$, in which rescue team $A(q)$ intersects the obstacle region, $\mathbb{F}^t$. Given that both $\mathbb{F}^t$ and $A(q)$ are closed sets in $\mathcal{W}$, the obstacle region, $\mathcal{C}_{obs}^t$, is also a closed set in $\mathcal{C}$.

The remaining configuration then becomes the time-varying path search space, $\mathcal{C}_{search}^t$, which is defined as:

$$\mathcal{C}_{search}^t = \mathcal{C} / \mathcal{C}_{obs}^t = \{q | q \in \mathcal{C}, \text{and } q \notin \mathcal{C}_{obs}^t\} \tag{10}$$

$\mathcal{C}_{obs}^{t}$ is a closed set in $\mathcal{C}$; therefore, $\mathcal{C}_{search}^{t}$ is required to be an open set, which implies that the rescue team can reach arbitrarily close to the obstacle, which is not practical in forest fire rescue scenarios. Thus, $\mathcal{C}_{search}^{t}$ needs to be redefined in a closed set approach [19]:

$$\mathcal{C}_{search}^{t} = cl(\mathcal{C}_{search}^{t}) = int(\mathcal{C}_{search}^{t}) \cup \partial\mathcal{C}_{search}^{t} \tag{11}$$

where $int(\mathcal{C}_{search}^{t})$ denotes the set of all points in $\mathcal{C}_{search}^{t}$ and $\partial\mathcal{C}_{search}^{t}$ denotes the set of all boundary points of $\mathcal{C}_{search}^{t}$.

### 2.4. Vertical Partitioning and Simplification of Time-Varying Search Space

Once $\mathcal{C}_{search}^{t}$ has been determined as the path search space, to decrease the computational burden, a technique known as vertical dissection [20] is employed to streamline $\mathcal{C}_{search}^{t}$. The process is outlined below:

First, split $\mathcal{C}_{search}^{t}$ into two finite families. The first finite family is cell cavity 1 and the second finite family is cell cavity 2. Each cell cavity 2 is a trapezoid or triangle with vertical edges. Dissect cell cavity 2 as follows: define $P$ as the set of vertices used to outline $\mathcal{C}_{obs}^{t}$, and let each vertex $p \in P$. From this point, extend a ray upwards and downwards through through $\mathcal{C}_{search}^{t}$ with the vertex as the endpoint until it reaches $\mathcal{C}_{obs}^{t}$. Depending on whether or not this ray can extend in both directions, four cases arise, which are illustrated in Figure 2.

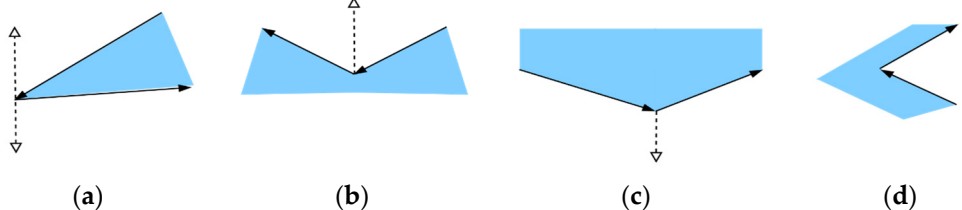

|     (a)     |     (b)     |     (c)     |     (d)     |

**Figure 2.** Four general cases of vertical sectioning: (**a**) can be extended upward and downward; (**b**) can only be extended upward; (**c**) can only be extended downward; (**d**) cannot be extended.

Splitting $\mathcal{C}_{obs}^{t}$ along the above rays produces a vertical dissection, and extending these rays produces a dissection of $\mathcal{C}_{search}^{t}$, as shown in Figure 3. The cell cavities 2 produced by this method can only include trapezoids and triangles. Each cell cavity 1 is a vertical line segment which is the boundary between two cell cavities 2.

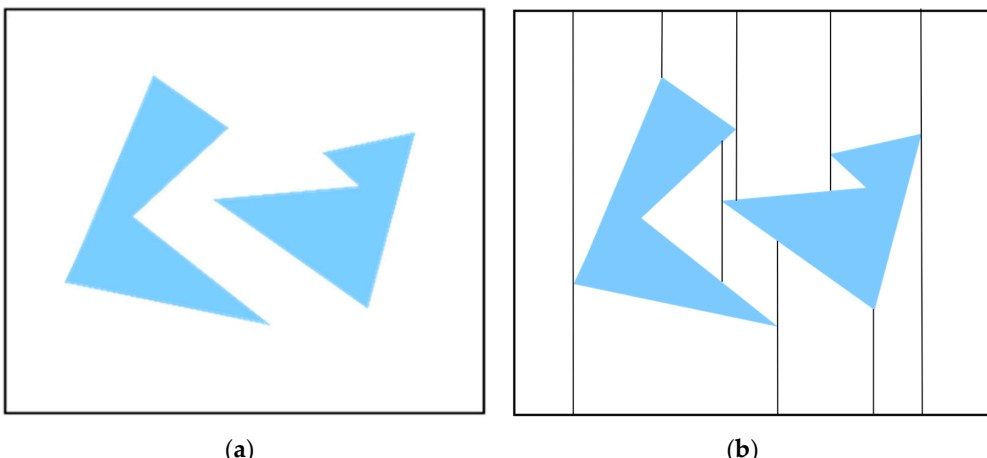

|     (a)     |     (b)     |

**Figure 3.** Vertical sectioning effect: (**a**) before sectioning; (**b**) after sectioning.

In order to accurately depict the topology of $\mathcal{C}_{search}^{t}$ following vertical dissection, cell cavity 2 is further explicitly defined as an open set on $\mathbb{R}^{2}$, indicating the interior of a system or triangle, and cell cavity 1 as the interior portion of a line segment.

After vertical dissection processing, a topology graph, $\mathcal{G}(V, E)$, needs to be defined in dealing with the motion planning problem in the cell cavity.

For each cavity $C_i$, let $q_i$ be denoted as a sample point. $q_i$ is the mass center of $C_i$ and $q_i \in V$ in the topology graph $\mathcal{G}(V, E)$. Then, there exists a mass center point for each cavity 1 and cavity 2. For cavity 1, its sampling point is its midpoint. For cavity 2, the sample point is represented by the centroid of the convex polygon. By connecting the sample points from neighboring cavity 1 to the corresponding sample points of cavity 2, the connection edges between the samples become the paths between the cell cavities. All the paths between adjacent cavities are connected to define the path topology graph, $\mathcal{G}(V, E)$. Each cell cavity 2 is a convex polygon; the route topology graph made by linking the mass centers can access each sample point, and therefore satisfies the accessibility condition, as well as the connectivity condition. $\mathcal{G}(V, E)$ is created from the cell cavity dissection; therefore, this dissection maintains the connectivity of $\mathcal{C}_{search}^t$. After obtaining the route topology graph, $\mathcal{G}(V, E)$, solve the path generation problem from the initial point $q_I$ to the target point $q_G$ further. Let $C_0$ and $C_k$ represent the cell cavities containing $q_I$ and $q_G$, respectively. Search a roadmap from $q_I$ to $q_G$ in the graph $\mathcal{G}(V, E)$. If it does not exist, the path is reported as unsolved. If it exists, let $C_1, C_2, \ldots, C_{k-1}$ represent the sequence of paths computed along from $C_0$ to $C_k$. The sample points $q_i (i = 1, 2, \ldots, k-1)$ in the cell $C_1, C_2, \ldots, C_{k-1}$ connect to provide an initial solution to the path planning. Using Equation (5), set $\tau(0) = q_I$ and $\tau(1) = q_G$. The path solution $\tau : [0, 1] \rightarrow \mathcal{C}_{search}^t$ can be obtained by connecting each point in $q_0$ to $q_k$ along the route topology graph. To ensure that the generated path solution does not conflict with obstacles, Figure 4 of the dissection process must be consulted.

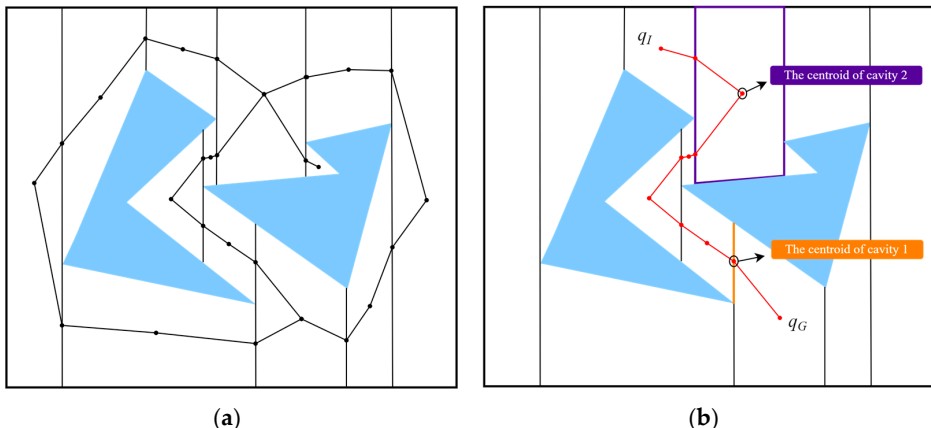

**(a)**                                                            **(b)**

**Figure 4.** Vertical dissection process: (**a**) topology of the dissected route; (**b**) initial solution of the path.

### 2.5. Improved Particle Swarm Optimization Search

These initial solutions are able to meet the needs of avoiding the forest fire spread area, but do not consider the terrain slope and travel speed as the path cost. Therefore, iterative optimization of the initial solutions in the path search space is required. In this study, a particle swarm algorithm was chosen to carry out the iterative optimization of the initial solutions, where the objective function of the optimization is a multi-element coupled path cost function of:

$$G(q_1, q_2, \ldots, q_k) = \frac{1}{\alpha \sum_{i=1}^{k} \frac{S(q_I, q_{i+1})}{\bar{V}(q_I, q_{i+1})} + \beta \sum_{i=1}^{k} mM(q_I, q_{i+1})} \tag{12}$$

where $q_i$ ($i = 1, 2, \ldots, k$) is each mass center passed by the target planning path, $S(q_i, q_{i+1})$, is the path length function from mass center $q_i$ to mass center $q_{i+1}$, and $\bar{V}(q_i, q_{i+1})$ is the velocity adjustment function determined by different slope magnitudes. The values of the speed adjustment function are detailed in Table 1. $M(q_i, q_{i+1})$ is the judgment function of whether the path falls into the preference region, and $m$ is the corresponding preference

coefficient. Thus, it is known that the preference coefficient for dangerous terrain areas is relatively low, whereas the preference coefficient for flat and safe areas is relatively high. $\alpha$ and $\beta$ are cost adjustment weights, which can be taken as 1. $g(q_1, q_2, \ldots, q_k)$ is the adaptation degree of the target planning path.

**Table 1.** Comparison table of walking speeds under different slope conditions.

| Slope (°) | Average Walking Speed (Steps/minute) | Average Walking Speed (km/h) | |
|---|---|---|---|
| | | Uphill | Downhill |
| 0~3 | 120 | 5.0 | 5.0 |
| 3~5 | 100 | 4.0 | 4.5 |
| 5~10 | 90 | 3.5 | 4.5 |
| 10~15 | 80 | 3.0 | 4.0 |
| 15~20 | 60~70 | 2.5 | 3.5 |
| 20~25 | 50~60 | 2.0 | 3.0 |
| 25~30 | 40~50 | 1.5 | 2.5 |

The original particle swarm algorithm [21] updates the velocity and direction of the search by sharing information among particles. Its key formula for particle update velocity is as follows:

$$v_{id}^{k+1} = \omega v^k + c_1 r_1 \left( p_{id,pbest}^k - x_{id}^k \right) + c_2 r_2 \left( p_{d,gbest}^k - x_{id}^k \right) \tag{13}$$

where $i$ denotes the particle ordinal number, $i = 1, 2, \ldots, N$; $d$ denotes the particle dimensional ordinal number, $d = 1, 2, \ldots, D$; $k$ denotes the number of iterations; $\omega$ denotes the inertia weight; $c_1$ and $c_2$ represent the individual learning factor and the social learning factor, respectively; $r_1$ and $r_2$ are random numbers in the interval [0, 1] to increase the search randomness; $v_{id}^k$ denotes the velocity vector of particle $i$ in the $d$th dimension in the $k$th iteration, $x_{id}^k$ denotes the position vector of particle $i$ in the $d$th dimension in the $k$th iteration, $p_{id,pbest}^k$ represents the historical optimal position of particle $i$ in the $d$th dimension in the $k$th iteration, i.e., the optimal solution obtained by the search of the $i$th particle (individual) after the $k$th iteration; $p_{d,gbest}^k$ represents the historical optimal position of the particle swarm in the $d$th dimension in the $k$th iteration, i.e., the optimal solution in the whole particle swarm after the $k$th iteration. In the original particle swarm optimization algorithm, the inertia weight, $\omega$, and learning factors, $c_1$ and $c_2$, are often set to constant values ($\omega = 1$, $c_1 = c_2 = 2$).

The previous particle swarm algorithm frequently results in poor route planning and global convergence due to the rapid attainment of local convergence. Therefore, this paper proposes a new time-varying velocity update approach to reduce the possibility of path planning in forest fire scenarios falling into local optimal solutions. The inertia weights, individual learning factors, and social learning factors in the speed update formulation are defined as follows:

$$\omega^{k+1} = \left[ w^k - \left( w^k - w_{min} \right) \right] \frac{r}{R} \tag{14}$$

$$c_1^{k+1} = \left[ c_1^k - \left( c_1^k - c_{1min} \right) \right] \frac{r}{R} \tag{15}$$

$$c_2^{k+1} = \left[ c_2^k - \left( c_2^k - c_{2min} \right) \right] \left( 1 - \frac{r}{R} \right) \tag{16}$$

The proposed approach aims to strike a balance between exploration and exploitation of the search space to achieve efficient and effective path planning in forest fire scenarios. From Equations (14)–(16), it can be seen that at the beginning of the iteration, the velocity update weight, $w^k$, is relatively large, the self-awareness factor, $c_1^k$, of the particle is relatively large, and the social awareness factor $c_2^k$ is relatively small, at which time the particle has a better global search ability. As the number of iterations increases, the velocity update

weight, $w^k$, decreases, the individual learning factor, $c_1^k$, decreases, and the social learning factor, $c_2^k$, increases, at which time the particle local search ability is enhanced.

## 3. Results and Discussion

To evaluate the efficacy and rationality of the designed algorithm, this study used DEM data and fire propagation data obtained through computational simulations at different times for a forest in Guangdong, China. The algorithm's performance was simulated with predetermined starting and endpoint inputs, and the calculated routes were obtained for different stages of fire propagation. Table 2 provides accuracy and projection details for the DEM data. Table 3 presents a summary of simulated particle swarm parameters and initial point parameters.

**Table 2.** Information of DEM data used in simulation.

| Properties | Description | Value |
|---|---|---|
| Raster Interpretation | Geometric nature of the raster | 'cells' |
| XIntrinsicLimits | Raster limits in intrinsic x coordinates | [0.5, 3612.5] |
| YIntrinsicLimits | Raster limits in intrinsic y coordinates | [0.5, 3612.5] |
| CellExtentInLatitude | Extent in latitude of individual cells | $2.7778 \times 10^{-4}$ |
| CellExtentInLongitude | Extent in longitude of individual cells | $2.7778 \times 10^{-4}$ |
| LatitudeLimits | Latitude limits of the geographic quadrangle bounding the georeferenced raster | [22.645233, 23.576067] |
| LongitudeLimits | Longitude limits of the geographic quadrangle bounding the georeferenced raster | [112.384655, 113.387988] |
| RasterSize | Number of rows and columns of the raster or image associated with the referencing object | [3351, 3612] |
| AngleUnit | Unit of measurement used for angle-valued properties | 'degree' |
| ColumnsStartFrom | Edge from which column indexing starts | 'north' |
| RowsStartFrom | Edge from which row indexing starts | 'west' |
| Coordinate System Type | Geographic coordinate reference system | 'geographic' |

**Table 3.** Parameters for simulation calculation.

| Case | Start Point | Target Point | N | D | $w_{min}$ | $[c_{1min}, c_{2min}]$ | Stage |
|---|---|---|---|---|---|---|---|
| 1 | 22°43′0.08″ N 112°36′32.79″ E | 22°43′34.86″ N 112°37′14.88″ E | 100 200 300 | 10 | 0.4 | [0.4, 0.4] | 1, 2, 3, 4, 5, 6 |
| 2 | 22°43′0.08″ N 112°36′32.79″ E | 22°43′34.86″ N 112°37′14.88″ E | 200 | 5 10 15 | 0.4 | [0.4, 0.4] | 1, 2, 3, 4, 5, 6 |
| 3 | 22°43′0.08″ N 112°36′32.79″ E | 22°43′34.86″ N 112°37′14.88″ E | 200 | 10 | 0.4 | [0.4, 0.4] [0.8, 0.8] [0.12, 0.12] | 1, 2, 3, 4, 5, 6 |

According to the computational experiments in [22], all initial values of $c_1$ and $c_2$ are set to 2, and the initial value of w is set to 0.9 to ensure that the initialized population can obtain an accurate solution. As stated in reference [23], when $c_1 + c_2 > 4$, it can lead to an increase in particle oscillations, and ultimately, the particles may range beyond the search space. When $c_1 + c_2 < 4$, the particles exhibit bounded periodic oscillations, resulting in better search performance.

After calculation, Figure 5 shows the planned paths for different stages of fire spread in Case1 when N = 100. It can be seen that the paths calculated by the algorithm avoid the fire grids on the map and reach the target point, achieving a good bypass effect. Figure 6a presents the average slope of the paths planned by the original PSO and the improved PSO at different stages of fire spread under the parameters of N = 100. It can clearly be seen that in stages 1 to 6, the slope of the improved PSO is smaller than that of the original PSO, indicating better performance. Figure 6b shows the maximum fitness values of the two algorithms corresponding to different stages of fire spread (a higher value indicates less action time and gentler slope). It can be seen from the figure that in stages 1, 3, 4, 5, and 6, the fitness value of the paths planned by the improved PSO is higher than that of the original PSO, whereas in stage 2, the fitness values of the two are almost the same.

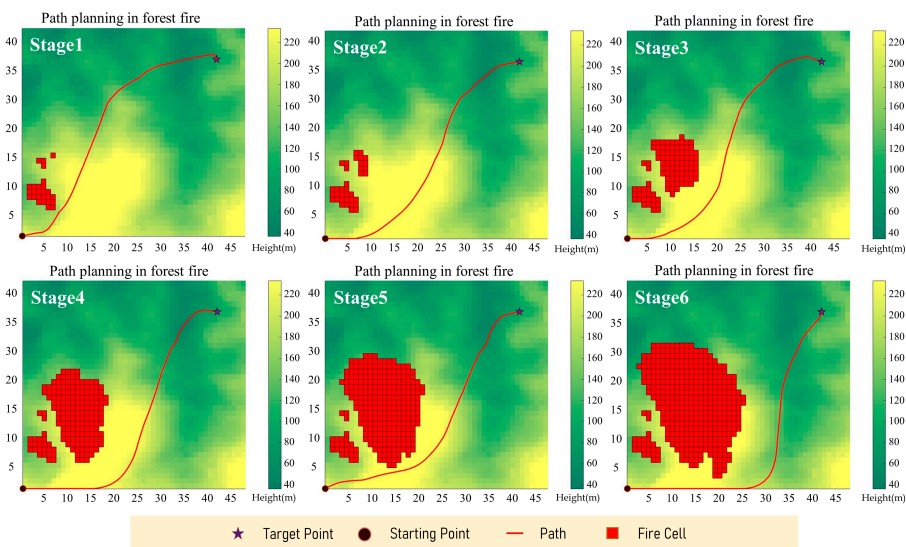

**Figure 5.** Schematic diagram of forest fire path planning based on improved particle swarm optimization (Case 1, N = 100).

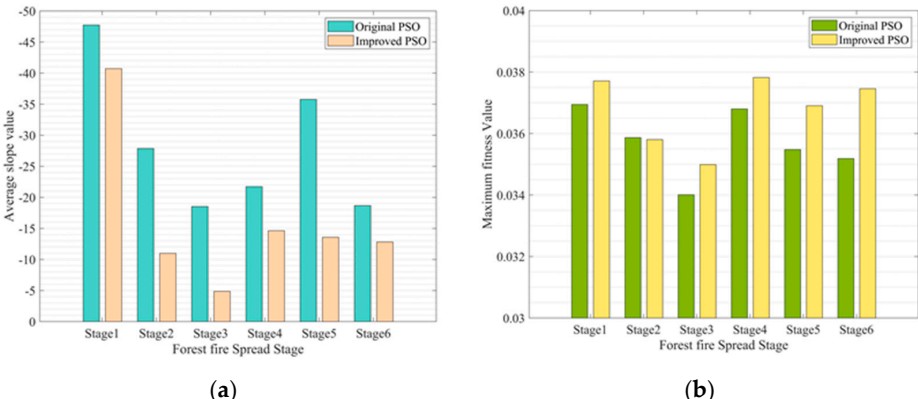

**Figure 6.** Comparison of path planning effects between the original PSO and improved PSO (Case 1, N = 100): (**a**) average slope of path; (**b**) maximum fitness value of path.

Furthermore, a–d illustrate the variations in the final path length and average slope of planned paths, with different particle numbers, N, particle dimensions, D, and minimum cognition factors.

In Figure 7a, it can be observed that with the increase in the number of particles, the total length of the final path showed a varying degree of reduction in most fire spread stages. When N = 100, the range of variation in the path length from Stage 1 to Stage 6 was approximately 1500~1900 m, and when N = 300, the range of variation in the path length from Stage 1 to Stage 6 became 1400~1600 m, indicating that the increase in the number of particles helps to find a shorter path during the planning process. In Figure 7b, due to the starting point being at a higher elevation on the map and the destination being at a lower elevation, the average slope of the obtained path is negative, with most of the slope variation ranging from $[-45°]$ to $[0°]$. Similarly, as the objective function takes into account the slope factor, with an increase in the number of particles, the optimization process tends to converge towards paths with smaller slopes.

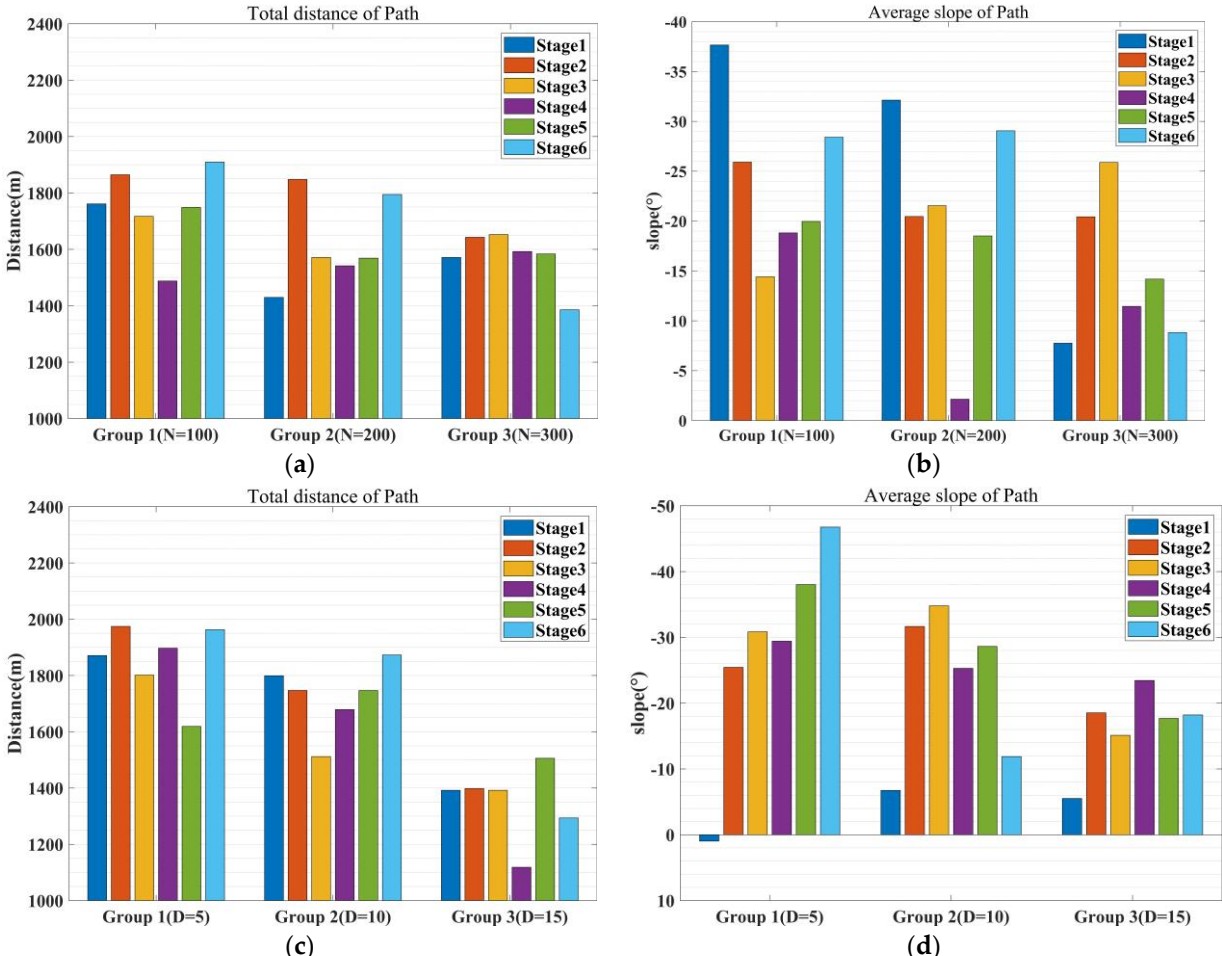

**Figure 7.** The variation in features under different particle dimensions and quantities. (**a**) The variations in total path length under different particle numbers. (**b**) The variations in average slope under different particle numbers. (**c**) The variation in total path length under different particle dimensions. (**d**) The variation in average slope under different particle dimensions.

In Figure 7c, the range of changes in path length is 1600~2000 m when D = 5, whereas it increases to approximately 1100~1500 m when D = 15. The increase in particle dimensionality actually further leads to an increase in the final path length. This suggests that particle dimensionality has a certain inhibitory effect on path length optimization, possibly due to the excessive number of path nodes leading to a more tortuous path. Additionally, in the process of optimization based on the objective function, the path length is not the only factor considered. When particle dimensionality is low, the uncertainty in slope changes between path nodes is greater, leading to them being trapped in a locally convergent state. This reason can be verified in Figure 7d, which shows that under a condition of particle dimensionality equal to 5, there is a positive average slope. This is because the existence of path nodes ranges from low to high, while in the simulated planning scenario, it is a process that ranges from high to low. Therefore, such solutions are generally difficult to meet the requirements of the path planning objective. In this application scenario, while ensuring the algorithm's time efficiency, a particle node number of 10 is a more reasonable choice.

In addition, this study tested the minimum values of the learning factors $c_{1min}$ and $c_{2min}$, with the aim of exploring the convergence of the algorithm under different learning factor conditions. Figure 8 shows the fitness variation during stage 3 of the propagation phase in Case 3.

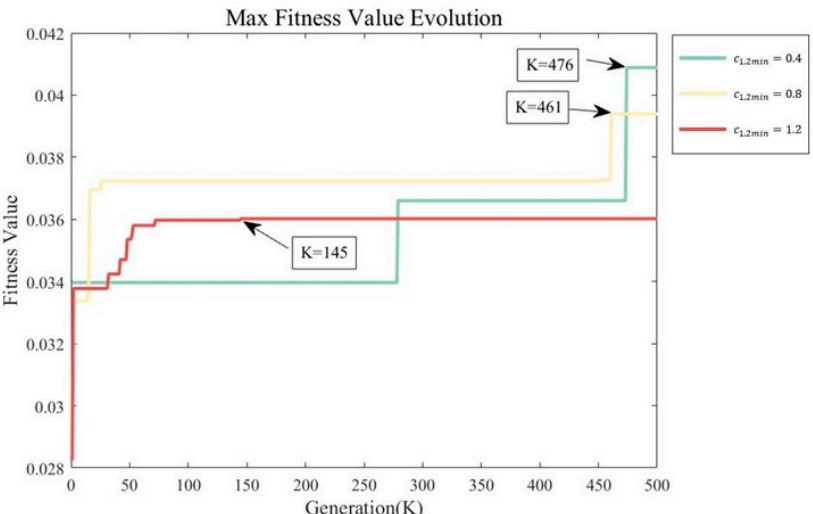

**Figure 8.** The convergence of path planning fitness under different learning factor conditions.

As shown in Figure 8, with other parameters held constant, increasing the learning factor results in a gradual decrease in the stable convergence generation of the particle swarm-based path planning algorithm. When both $c_{1min}$ and $c_{2min}$ are 0.4, the particle swarm converges in 476 generations. When both $c_{1min}$ and $c_{2min}$ are 0.8, the particle swarm converges in 461 generations. When both $c_{1min}$ and $c_{2min}$ are 1.2, the particle swarm converges in 145 generations. Hence, the variation in the minimum learning factor in the particle swarm has a notable impact on the number of the generations required for convergence. Moreover, as the learning factor increases, the number of generations required to reach convergence becomes shorter; however, this may not always lead to obtaining the optimal solution. Figure 8 highlights that the path planning algorithm's fitness value at convergence is the highest when the minimum learning factor is 0.4, compared with the lowest value obtained when the minimum learning factor is 1.2. Hence, the minimum learning factor in this model needs to be determined according to the actual computational scenario.

## 4. Conclusions

In summary, this paper illustrates how to conduct path planning considering multiple risk factors and costs through the use of forest fire scenarios. By modeling the configuration space of the fire rescue team and the spatiotemporal space of forest fire propagation, we established a time-varying path search space. We simplified the search space through vertical cell decomposition, reducing the complexity of search space in the forest fire path planning scenario. In addition, we established a path planning model based on particle swarm optimization which optimized path length and path slope. Furthermore, the path planned by the algorithm exhibited an excellent "detour effect" in the dynamic spread of the fire while still achieving its objective of reaching the target point.

We evaluated the planning performance of the original particle swarm optimization algorithm and the improved particle swarm optimization algorithm under the same parameter settings. The results showed that optimizing the particle swarm optimization algorithm led to paths with smaller slope and higher fitness values at different stages of fire spread.

Moreover, during the simulation process, by setting different cases, we found that the number of particles and the dimensions of the particles in the particle swarm optimization algorithm can lead to changes in the final convergence and results. Increasing particle dimensions provide more reasonable and stable routes for the model while decreasing the convergence iterations.

Our research shows that the improved particle swarm optimization algorithm can adapt well to the path planning optimization task in forest fire spread scenarios. In future

work, we plan to optimize the particle swarm parameters and objective function model and explore the impact of detailed parameter changes in the improved particle swarm model on optimization results. We will also incorporate more risk factors into the objective function of this algorithm to achieve a more reasonable path planning effect.

**Author Contributions:** Conceptualization, K.L. and L.Z.; methodology, K.L. and Z.F.; software, K.L.; validation, K.L., L.Z. and L.H.; formal analysis, K.L.; investigation, L.H.; resources, L.H.; data curation, L.Z.; writing—original draft preparation, K.L. and L.Z.; writing—review and editing, L.H.; visualization, K.L.; supervision, T.C.; project administration, T.C.; funding acquisition, T.C. All authors have read and agreed to the published version of the manuscript.

**Funding:** This study was supported by National Key R&D Program of China (Grant No. 2022YFC3003103), National Natural Science Foundation of China, China (Grant No. 72104123).

**Institutional Review Board Statement:** Not applicable.

**Informed Consent Statement:** Not applicable.

**Data Availability Statement:** Data sets used during the current study are available from the corresponding author on reasonable request. https://www.gscloud.cn/sources/index?pid=302 (accessed on 5 April 2023).

**Acknowledgments:** This work is supported by the study on dynamic fire spread prediction technology in the forest town border region (Grant No. 2022YFC3003103) and the study on epidemic prevention and control methods considering the aggregation behavior of large event crowds.

**Conflicts of Interest:** The authors declare no conflict of interest.

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
