# Peer review of "Improved Particle Swarm Path Planning Algorithm with Multi-Factor Coupling in Forest Fire Spread Scenarios"

_fire, doi:10.3390/fire6050202_

Round 1

Reviewer 1 Report

This article achieves path planning in the dynamic spread of forest fires by improving the classic particle swarm theory and using topology modeling methods. The effectiveness of the planned paths is also evaluated to determine the appropriate model parameters for the computational scene. This has some application value in rescue and escape in actual forest fire scenarios, but there are still some problems:

  1. Why is the particle swarm algorithm used as the basis for path planning rather than other heuristic algorithms? Is there any special consideration or advantage in using this algorithm compared to other heuristic algorithms in the context of forest fire spread? Please explain the advantages and necessity of using the improved particle swarm method in this article.
  2. Are the inertia weights, individual learning factors, and group learning factors of the particle swarm model used in Table 3 determined based on any theoretical basis or reference? It is recommended to clarify the reference source for these three numbers, otherwise readers may believe that the selection of these values is subjective and without scientific basis.
  3. The analysis in Figure 7 can be described using accurate data, especially when authors use terms such as "increase" or "decrease" without accompanying precise numbers. This makes the authors' reading less rigorous.

In summary, if authors consider publishing further articles in this journal, they should modify and resubmit.

English need to be improved

Reviewer 2 Report

This paper is for path planning in the case of forest fires without road networks, the proposed improved particle swarm path planning algorithm based on can design a good path around the fire area, the paper got a good experimental effect, the paper as a whole is clear and smooth, but the paper still has some problems :

1. The paper only validates the proposed method, it is suggested to add some comparison tests, such as the particle swarm algorithm without improvement and the algorithm proposed in this paper, what is the advantage of the algorithm proposed in this paper in path planning?

2. Why only the effect of slope on speed is considered, does wind, altitude, etc. (just for example) also have some effect on speed?

3. Does Height in Figure 5 refer to the altitude?

4. q C or q ? in Equation 9 and in line 136

5. The path between the sample points of the cavity is observed in Figure 4 as connecting the edges before connecting to the sample points, how are the points on the edges determined?

6. Line 171 "For cavity 2, define an edge from its sample point along its boundary for each sample point of cavity 1." is not clear enough.

7. Is the N in Figure 5 the number of masses?

8. Figure 5 is suggested to enlarge the font some more, the font is a little blurred.
